# Gene Signatures of Symptomatic and Asymptomatic Clinical-Immunological Profiles of Human Infection by *Leishmania* (*L.*) *chagasi* in Amazonian Brazil

**DOI:** 10.3390/microorganisms11030653

**Published:** 2023-03-03

**Authors:** Vania Lucia R. da Matta, André N. Gonçalves, Cláudia Maria C. Gomes, Islam H. Chouman, Frederico M. Ferreira, Marliane B. Campos, Luciana V. Lima, Thiago Vasconcelos dos Santos, Patrícia Karla Ramos, Rodrigo R. Furtado, Marcia D. Laurenti, Carlos Eduardo P. Corbett, Helder I. Nakaya, Fernando T. Silveira

**Affiliations:** 1Laboratorio de Patologia de Molestias Infecciosas (LIM-50), Departamento de Patologia, Faculdade de Medicina, Universidade de Sao Paulo, Sao Paulo 01246 903, Brazil; 2Parasitology Department, Evandro Chagas Institute, Health Surveillance Secretary, Ministry of Health, Ananindeua 67030-000, Brazil; 3Departamento de Análises Clínicas e Toxicológicas, Faculdade de Ciencias Farmaceuticas, Universidade de Sao Paulo, Sao Paulo 05508-220, Brazil; 4Hospital Israelita Albert Einstein, Sao Paulo 05652-900, Brazil; 5Tropical Medicine Nucleus, Pará Federal University, Belém 67030-000, Brazil

**Keywords:** *Leishmania* (*L.*) *chagasi*, human infection, transcriptomic analysis, symptomatic, asymptomatic, clinical-immunological profiles

## Abstract

Individuals infected with *Leishmania* (*L.*) *chagasi* may present different asymptomatic and symptomatic stages of infection, which vary in the clinical–immunological profiles that can be classified as asymptomatic infection (AI), subclinical resistant infection (SRI), indeterminate initial infection (III), subclinical oligosymptomatic infection (SOI), and symptomatic infection (SI) (=American visceral leishmaniasis, AVL). However, little is known about the molecular differences between individuals having each profile. Here, we performed whole-blood transcriptomic analyses of 56 infected individuals from Pará State (Brazilian Amazon), covering all five profiles. We then identified the gene signatures of each profile by comparing their transcriptome with those of 11 healthy individuals from the same area. Symptomatic individuals with SI (=AVL) and SOI profiles showed higher transcriptome perturbation when compared to those asymptomatic III, AI and SRI profiles, suggesting that disease severity may be associated with greater transcriptomic changes. Although the expression of many genes was altered on each profile, very few genes were shared among the profiles. This indicated that each profile has a unique gene signature. The innate immune system pathway was strongly activated only in asymptomatic AI and SRI profiles, suggesting the control of infection. In turn, pathways such as MHC Class II antigen presentation and NF-kB activation in B cells seemed to be specifically induced in symptomatic SI (=AVL) and SOI profiles. Moreover, cellular response to starvation was down-regulated in those symptomatic profiles. Overall, this study revealed five distinct transcriptional patterns associated to the clinical–immunological (symptomatic and asymptomatic) profiles of human *L.* (*L.*) *chagasi*-infection in the Brazilian Amazon.

## 1. Introduction

Human infection with *Leishmania* (*L.*) *chagasi* Lainson and Shaw 1987 (= *Leishmania chagasi* Cunha and Chagas 1937) [1,2,3] may cause American visceral leishmaniasis (AVL), also known as “neotropical calazar” [4,5]. Individuals with AVL often have daily fever lasting up to two months, weakness, asthenia, loss of appetite, weight loss, skin-mucous pallor, diarrhea, abdominal distension, and hepatosplenomegaly [6,7]. However, depending on the genetic factors and immune response of the individuals, the manifestation of the disease can vary considerably, ranging from an asymptomatic infection to a severe infection (AVL) that can lead to death [8,9].

In the Brazilian Amazon, it has been demonstrated that individuals infected with *L.* (*L.*) *chagasi* can be classified into five profiles within a clinical–immunological spectrum [10]. This classification is based on the clinical signs and symptoms of infected individuals, as well as their cellular and humoral immune responses to the infection, as measured by the delayed-type hypersensitivity reaction (DTH) and indirect fluorescence antibody test (IFAT-IgG) to species-specific *L.* (*L.*) *chagasi* antigens. Initially, when the immune system is responding to the infection and the infection outcome is not yet determined, i.e., whether resistant or susceptible, infected individuals will display a clinical–immunological profile known as indeterminate initial infection (III = DTH^−^/IFAT^+/++^). Following its evolution, the infection may then evolve to resistant asymptomatic or to susceptible symptomatic clinical–immunological profiles. While asymptomatic clinical–immunological profiles can be classified into asymptomatic infection (AI = DTH^++++^/IFAT^−^) and subclinical resistant infection (SRI = DTH^++++^/IFAT^++^), symptomatic ones can be classified into subclinical oligosymptomatic infection (SOI) and symptomatic infection (SI = AVL), both with the same immune response (DTH^−^/IFAT^++++^) [11,12,13,14].

Recently, a molecular diagnostic approach performed in the Brazilian Amazon corroborated the clinical–immunological spectrum of the infection previously described based on DTH/IFAT assays, recognizing the five clinical–immunological infection profiles through a qPCR analysis of the urine of infected individuals [15]. Additionally, it has also been observed that each profile shows distinct levels of pro-inflammatory (TNF-α and IFN-γ) and regulatory (IL-6 and IL-10) cytokines [16].

Although few studies have characterized the transcriptomic changes between AI and SI (=AVL) profiles [17,18], little is known about the molecular differences among these five clinical–immunological infection profiles. Here, we provided a comprehensive overview of the transcriptomic changes in the blood of individuals infected with *L.* (*L.*) *chagasi*, covering all the five profiles. Our findings shed some light on genes and pathways potentially involved with the progression of the disease or the resistance to it.

## 2. Material and Methods

### 2.1. Study Area

The present study was carried out in rural area of Bujaru municipality, in northeastern Pará State (01°30′54″ S: 48°02′41″ W), in the Brazilian Amazon. The climate is typically equatorial, with a mean temperature of 28 °C and high humidity. The municipality of Bujaru has lived with AVL for more than 30 years, when the disease strongly expanded into northern and northeastern Pará State. During the most recent six-year period (2012–2017), the average number/year of AVL cases in Bujaru was 7.8 [19].

### 2.2. Population and Study Design

This was a cross-sectional work (2018–2019) that analyzed the whole blood gene expression of individuals living in rural area of Bujaru municipality (Pará State), in the Brazilian Amazon. Written informed consent was obtained from individuals or their guardians, and the study was approved by the Ethics Committee of the Medicine School of São Paulo University, São Paulo State, Brazil (number: 62660716.7.0000.0065).

To identify infected, symptomatic, and asymptomatic individuals, in the endemic area, an infection screening was performed based on the use of species-specific DTH and IFAT-IgG assays. Amastigotes of *L.* (*L.*) *chagasi* isolated from the studied region were used to perform the IFAT-IgG test, and promastigotes of the same isolate were processed and fixed in merthiolate solution (1:10,000) to be used in the DTH assay [10,11,12]. Individuals who showed reactivity to either DTH or IFAT-IgG or both were clinically evaluated. Among the screened individuals, we randomly selected 56 cases with the following distribution: 15 AI (6 males, average age 16.6), 9 SRI (5 males, average age 16.2), 19 III (8 males, average age 19), 3 SOI (1 male, average age 25), and 10 SI (=AVL) (4 males, average age 32.6). In addition, 11 healthy individuals (6 males, average age 11.9) from the same area were included as control group.

### 2.3. Clinical–Immunological Profiles

The criteria for discriminating cases of infection and clinical–immunological profiles, as well as susceptible or resistant cases, have been described in previous studies [10,11,12,14]. Briefly, infection cases were identified by reactivity to DTH, IFAT-IgG, or both assays. AI profile was defined by DTH reactivity and absence of specific IgG-antibody response [below the IFAT-IgG dilution cut-off of 1:80], which is strongly linked to genetic resistance to infection [8]. The SI profile (=AVL) was defined by clear DTH (cellular immune response) inhibition and a high expression of specific IgG-antibody (humoral) response. The SOI and SRI profiles represent borderline genetic expressions of susceptibility and resistance to disease, respectively. The former generally evolves into mild clinical signs of susceptibility, with spontaneous resolution of clinical infection after one to two months [20]. The latter presents an asymptomatic stage often evolving to the AI resistance profile. Finally, individuals with profile III are usually asymptomatic and represent the earliest stage of infection but, in some cases, there have been diagnosed asymptomatic individuals with III profile presenting IgM serological reactivity that evolved towards to AVL following a short six-week incubation period [21,22].

We performed a complete clinical and physical examination on all infected individuals. The classical signs and/or symptoms of AVL were daily fever lasting up to two months, weakness, asthenia, loss of appetite, weight loss, skin–mucous pallor, diarrhea, abdominal distension, and hepatosplenomegaly, as well as hematological changes, such as anemia, leukopenia, and thrombocytopenia. Individuals with SOI profile often have fever, asthenia, cutaneous pallor, and moderate enlargement of the spleen [5,6,13,20]. Only cases presenting typical features of AVL received conventional antimony therapy, as recommended by the Brazilian AVL Control Program [23]. SOI profile cases with mild or moderate symptoms were monitored fortnightly for a period of two months through clinical (with a complete physical examination) and laboratory (IFAT-IgG assay and blood count) evaluations [20]. All individuals with profile III were examined weekly for periods of up to three months, respecting the probable incubation period of SI profile (=AVL) [5,6]. They also underwent serological IgM-antibody screening (IFAT-IgM), as profile III cases with IgM reactivity have been shown to develop AVL (=SI profile), indicating that IgM serological reactivity in asymptomatic III profile cases appears to be a strong marker AVL development [21,22,24].

### 2.4. Blood Transcriptomics and Data Analysis

Peripheral blood was collected into PAXgene blood RNA tubes (Qiagen, Germantown MD, USA). Total RNA was isolated using PAXgene blood RNA Kit (PreAnalytiX, Cotati, CA, USA), according to the manufacturer’s instructions. Sample quality and integrity were assessed by the Tape Station 4200 system (Agilent Technologies, Santa Clara, CA, USA). cDNA libraries were prepared using TruSeq Stranded total RNA (Illumina, San Diego, CA, USA) with Ribo-Zero Globin magnetic beads to deplete globin-encoding mRNA in addition to the cytoplasmic and mitochondrial rRNA. The high-throughput sequencing of the transcriptome (RNA-seq) was performed in the HiSeq 2500 Illumina platform with V4 reagents. The quality of the RNA-seq data were assessed with the FASTQC program [25]. The sequenced reads were aligned to the human genome reference and counted with the program STAR [26]. The read counts, in turn, were submitted to batch effect correction with the R package LIMMA [27], following quantile normalization and differential expression analysis with the R package EdgeR [28]. Functional enrichment analyses were performed with EnrichR tool using Reactome 2022 database [29]. To assess whether the transcriptome of SI profile (=AVL) patients was more “perturbed” than the other profiles, we performed a molecular degree of perturbation (MDP) analysis using the healthy individuals as the reference group [30].

### 2.5. Ethical Considerations

Written informed consent was obtained from infected individuals or their guardians, and the study was authorized by the Ethics Committee of the Medicine School of São Paulo University, São Paulo State, Brazil (number: 62660716.7.0000.0065).

## 3. Results

We collected whole blood samples from 56 individuals infected with *L.* (*L.*) *chagasi* and 11 healthy non-infected controls from the same endemic area (Figure 1A). Based on their clinical–immunological profiles at the time of enrollment, we classified the infected individuals in five profiles: AI, SRI, III, SOI, and SI (=AVL) (Figure 1A). We then performed RNA sequencing of all 67 samples.

Unsupervised principal component analysis suggested that individuals with SI profile (=AVL) form a separated cluster (Figure 1B). Molecular Degree of Perturbation analysis (cite: https://www.ncbi.nlm.nih.gov/pmc/articles/PMC6822058/, accessed on 12 November 2022) revealed that individuals with SI (=AVL) and SOI profiles showed higher transcriptomic perturbation when compared to the other profiles (Figure 1C).

We then identified the differentially expressed genes (DEGs) whose expression was up- or down-regulated on each profile when compared to healthy controls. Subjects with SOI and SI (=AVL) profiles have 1849 and 1341 up-regulated genes, respectively; whereas individuals with AI and SRI profiles have 1148 and 1350 up-regulated genes, respectively (Figure 2A, Appendix A). Although the number of up-regulated genes on each of these 4 profiles was roughly the same (average of 1422 genes), very few of these up-regulated genes were shared among the profiles (Figure 2B). Similar results were found for down-regulated genes (Figure 2A,B).

We then performed pathway enrichment analysis using the 1006 genes and 2650 genes whose expression were specifically altered in either asymptomatic individuals (AI/SRI profiles) or symptomatic individuals (SOI/SI = AVL profiles), respectively (Figure 2C). *Toll-like* receptor cascades, pro-inflammatory response, and neutrophil degranulation, among others, were pathways enriched with genes specifically up-regulated in asymptomatic (AI/SRI) profiles (Figure 2C). Other immune-related pathways, such as MHC Class II antigen presentation, activation of NF-kB in B cells, and adaptive immune responses, seem to be specifically induced in symptomatic (SOI/SI = AVL) profiles. Conversely, cellular response to starvation and response of EIF2AK4 (GCN2) to amino acid deficiency are enriched in down-regulated genes in symptomatic (SOI/SI = AVL) profiles (Figure 2C).

Pathway enrichment analysis using genes up-regulated on each profile was also performed (Figure 3). This analysis differs from the previous one where we utilized genes exclusively up- or down-regulated in specific profiles (Figure 2C). By utilizing all up-regulated genes of each profile (Figure 2A), we captured signatures that were shared between profiles. Despite the fact that the pathways related to cell cycle and MHC Class II antigen presentation were only enriched in individuals with SI profile (=AVL), most pathways were enriched in individuals with AI and SI (=AVL) profiles (Figure 3). Moreover, several pathways were shared and activated at the same level between SOI and SRI profiles, both with intrinsic genetic characteristics.

We also found genes showing distinct transcriptional levels throughout the clinical–immunological spectrum of human infection by *L.* (*L.*) *chagasi* (Figure 4).

Regarding the follow-up of the cases of infection during the study period (2018–2019), it is important to mention that, after blood sample collection, none of the three individuals in the SOI profile, which were followed biweekly for a three-month period, evolved towards to AVL. On the contrary, all of them presented spontaneous remission of symptoms (weakness and/or general malaise, lack of appetite, and irregular moderate hyperthermia) thirty to forty-five days after the clinical and laboratory diagnosis. Similarly, none of the asymptomatic individuals with III profile, which were followed weekly for periods of up to three months, progressed to AVL. In fact, all III profile cases (with negative IgM serology) evolved into the AI profile with positive DTH conversion. All 10 patients with AVL received antimony therapy (15 mg/SB^v^/kg weight/25 days) as recommended by the Visceral Leishmaniasis Control Program of the Brazilian Ministry of Healthy, showing satisfactory clinical remission after antimony therapy.

## 4. Discussion

Understanding how human genetics can influence susceptibility or resistance to infectious diseases was the main reason for the present study, besides to gaining molecular evidence to subsidize the previous diagnostic approach of the clinical–immunological spectrum of human infection by *L.* (*L.*) *chagasi* Lainson and Shaw 1987, i.e., the causal agent of American visceral leishmaniasis (AVL) [1,2,3], based on the DTH and IFAT-IgG assays combined with the clinical picture of infected individuals [10]. As a result of this diagnostic approach, it was possible to identify five different clinical–immunological profiles of infection, three asymptomatic (III, SRI and AI) and two symptomatic (SI = AVL and SOI), among which it is important to draw attention for the III profile (Indeterminate Initial Infection) characterized by IgG-antibody reactivity alone (DTH^−^/IFAT^+/++^), from where the infection can migrate, depending on the immunogenetic character of the infected individual, to the resistance pole of infection represented by the AI profile (DTH^+/++++^/IFAT^−^), or for the susceptibility pole represented by the SI profile (=AVL) (DTH^−^/IFAT^+/++++^) [11,12,13,14]. Subsequently, other cohort studies performed in the Brazilian Amazon regarding on the dynamics of the clinical–immunological profiles of infection concluded that 1–3% of III profile cases can progress to AVL, which was confirmed in two situations when it was performed the preclinical diagnosis of AVL by demonstrating IgM-antibody response in two III asymptomatic individuals, six weeks before both individuals manifested the first signs and/or symptoms of AVL [21,22]. Taking in account these considerations, this represents the first study performed in the Brazilian Amazon that carried out transcriptomic analysis of human *L.* (*L.*) *chagasi*-infection with emphasis on the five different clinical–immunological profiles, symptomatic (SI = AVL and SOI) and asymptomatic (AI, SRI, and III), of that infection.

In the present study, we detected higher blood transcriptional perturbation in infected individuals with symptomatic SI (=AVL) and SOI profiles when compared to those with asymptomatic AI and SRI profiles, suggesting that infection and/or disease severity may be associated with higher transcriptomic changes. Of note, a high number of exclusive genes was seen in each profile (Figure 2A,B), indicating that each one has a unique gene signature. Among the pathways specifically associated with asymptomatic individuals (AI/SRI profiles), the strong activation of the innate immune system stands out (Figure 2C). Neutrophils are crucial elements of the innate immune system and play an important role in the immune response to *Leishmania* infection [31]. They are the innate first responder cells, which are armed with an arsenal of anti-parasitic activities that includes phagocytosis, the release of reactive oxygen and nitrogen species (ROS and RNS), formation of neutrophil extracellular traps (NETs), and the release of granule-derived toxic compounds in the local environment or into the phagosome, which compromise the integrity of the parasite membrane [32,33,34].

Here, one of the most up-regulated pathways found in asymptomatic infected individuals (AI/SRI profiles) was the neutrophil degranulation. Elastase is found in abundance in neutrophils and is stored in azurophil granules together with other proteins involved in anti-microbial defense [35]. After polymorphonuclear cell (PMN) exposure to *L.* (*L.*) *infantum*, a significant exocytosis of myeloperoxidase (MPO) and neutrophil elastase (NE) occurs, contributing to the parasite control [33,36]. Moreover, NE is also released with other granule proteins and chromatin to constitute PMN NETs [37] that entraps *L.* (*L.*) *infantum* promastigotes, causing a progressive loss of parasite viability [32,33]. Previous studies showed that NE also up-regulates inflammatory cytokine production by macrophage (IL-8, IL-1β, and TNFα) [35], and the implication of NE in the induction of cytokine expression involves the nuclear factor κB (NFκB) [38], and the activation of *toll-like* receptor 4 (*TLR*-4) [39,40].

Activation of the *toll-like* receptors (*TLR*s) cascade was also a specific up-regulated pathway found in asymptomatic infected individuals (AI/SRI profiles). *TLR*s are another hallmark of the innate immune response. They are mostly expressed by antigen-presenting cells (APCs), and their engagement with pathogen associated molecular patterns (*PAMPs*) on the *Leishmania* surface leads to the activation of the microbicidal responses of phagocytes, such as the production of superoxide and nitric oxide [41]. *TLR*2 is the most relevant to *Leishmania* infection and is centrally responsible for the recognition of lipophosphoglycan (LPG), the greatest expressed surface molecule in *Leishmania* parasites [42,43]. *TLR*2 is involved in the resistance against *L.* (*L.*) *infantum*-infection since it promotes the development of Th1 and Th17 protective immune responses, and also the activation of dendritic cells and neutrophils, which trigger the TNF-α and nitric oxide production, amplifying the immunity against *L.* (*L.*) *infantum* (and probably *L.* (*L.*) *chagasi* as well) [44,45]. Previous studies also demonstrated a role for *TLR*3 in *L.* (*L.*) *donovani* recognition, phagocytosis, and induction of leishmanicidal activity in macrophages [46]. In turn, *TLR*-4 expression enhances after *L.* (*L.*) *donovani* infection, and knockout *TLR*4 ^−/−^ infected mice exhibit reduced interferon-gamma (IFN-γ), tumor necrosis factor (TNF), and inducible nitric oxide synthase (iNOS) mRNA expression in the liver, showing its involvement in parasite restraint [47]. Additionally, *TLR*9 plays a critical role in neutrophil recruitment during the protective response against *L.* (*L.*) *infantum*-infection [48]. It also has been demonstrated that *L.* (*L.*) *infantum* (and probably *L.* (*L.*) *chagasi* as well) activates dendritic cells via *TLR*9 for the generation of IL-12, which subsequently triggers natural killer cells (NK), cytotoxicity, and IFN-γ production, the main cytokine that drives macrophage activation, a target cell for *Leishmania* parasites [49]. Collectively, several *TLR*s are involved in host protective mechanisms against *L.* (*L.*) *infantum*. However, *TLR* functions are diverse and may be related to severity or a protective response in a species-dependent manner [45]. In this sense, *TLR*9 is significantly more expressed in *Leishmania* (*L.*) *amazonensis* infections in severe anergic diffuse cutaneous leishmaniasis (ADCL) in comparison to localized cutaneous leishmaniasis (LCL) caused by the same leishmanine species [50], and *TLR*2 mediates non-protective immune response in *L.* (*L.*) *amazonensis* and *Leishmania* (*V.*) *braziliensis* infected mice, in contrast to *L.* (*L.*) *infantum*-infection [45]. Altogether, our findings suggest that the innate immunity through the activation of phagocytic cells, specially neutrophils, *TLR*s, and pro-inflammatory cytokines are the major characters that support protective immunity in the asymptomatic profiles with moderate (SRI) to strong resistance (AI) against *L.* (*L.*) *chagasi*-infection.

On the other hand, some prominent pathways specifically associated with symptomatic individuals (AVL/SOI profile) were those related to activation of NF-kappa B in B cells and cell cycle, and repression of the cellular response to starvation. The nuclear factor kappa light chain enhancer of activated B cells (NF-κB) is a small family of highly conserved transcription factors that control, in multiple ways, both innate and adaptive immune responses, as well as the development and maintenance of cells and tissues [38]. NF-κB activation in B cells predominately prevents apoptosis of B cells and plays a critical role in their development, survival, and activation [51,52,53]. On the other hand, plasma cells (PCs) represent the terminal differentiation step of mature B lymphocytes, and are the unique cells able to secrete antibodies, being a key component of the humoral immunity [54].

For decades now, hypergammaglobulinemia, besides a high production of specific and non-specific (polyclonal) antibodies, are considered hallmarks of human visceral leishmaniasis [55,56]. Here, besides a negative cellular immune response to the intradermal challenge with specific *L.* (*L.*) *chagasi* antigen (DTH^−^), all AVL patients presented high levels of specific IgG-antibody response (IFAT-IgG^++++^). In this sense, CD38, which is considered as one of the most reliable markers for the identification of PCs [57], together with XBP1, a transcription factor that governs plasma cell differentiation and its function, were exclusively up-regulated in our AVL patients [58]. Thus, we understand that the activation of NF-κB in our AVL patients plays a pivotal role in the full development, survival, and activation of B cells which, after B cell receptor (BCR) recognition of *Leishmania* antigens, differentiated into plasma cells under the XBP1 influence, favoring the high production of specific IgG-antibody response observed in the overt disease. Of note, the abundance of plasma cells in our AVL patients was significantly higher than that in other clinical–immunological infection profiles studied. These observations are consistent with the understanding that polyclonal activation of B cells and high IgG-antibody production contribute to parasite survival and disease exacerbation [59].

Unexpectedly, the cellular response to starvation pathway was down-regulated in symptomatic AVL and SOI profile patients (Figure 2C). Starvation, the most extreme form of malnutrition, is the result of a severe or total lack of nutrients needed for the maintenance of life. Deprivation of nutrients causes a wide range of effects on the body, influencing organ size, hormone production, and innate and acquired immune responses, and it has been long related to increased susceptibility to infectious diseases [60,61,62,63]. Decreased M1 macrophages, cytotoxic cells, and Th1 effector cells are observed in malnutrition conditions, impairing the protective immunity as a whole [63]. Visceral leishmaniasis (VL) mostly affects poverty populations, and malnutrition is both a risk for the development of VL and a consequence of the disease [60,64]. In this way, malnourished mice showed an impaired lymph node capacity to act as a barrier to the dissemination of *L.* (*L.*) *donovani*, early parasite visceralization, and decreased nitric oxide production following parasite infection [61,65]. Protein undernutrition also contributed to leukocyte depletion, favoring the development of VL, and influencing severity in *L.* (*L.*) *infantum* infected hamsters [66]. Moreover, after establishing malnutrition, mice inoculated with *L.* (*L.*) *infantum* presented significant decrease in liver and spleen weight, and increased parasite burden in these organs compared to the well-nourished control mice counterparts [67].

Leptin, an adipocytokine, regulates the balance between food intake and energy expenditure, and a low level of leptin has been considered a biomarker of malnutrition and/or starvation [68]. Such a molecule is also implicated in several immune regulation activities [69]. In this context, the addition of exogenous leptin in *L.* (*L.*) *donovani* infected human monocytic cell lines promotes a Th1 response through the up-regulation of pro-inflammatory cytokines and stimulates macrophages by enhancing the intracellular ROS generation, helping with the oxidative killing of parasites [70]. Malnourished mice infected with *L.* (*L.*) *donovani* showed significantly reduced levels of leptin, which worsened VL. However, exogenous administration of leptin induced Th1 polarization, diminished Th2 response, controlled parasite growth in visceral organs, and induced hepatic granulomatous response to clear the infection [71]. In human active VL caused by *L.* (*L.*) *infantum*, leptin levels were decreased in 50% of the patients. In those, the diminished leptin positively correlated with severity parameters, such as lower albumin and hemoglobin levels. After anti *Leishmania*-treatment, leptin increased to normal levels [72].

Starvation and malnutrition are both associated with poverty and reduced leptin levels. In our casuistic, the leptin expression in AVL patients was equal to the healthy controls. Interestingly, Harhay et al. [73] found substantial heterogeneity in the nutritional profile of VL patients in countries that compose about 90% of the world’s burden of VL, including Brazil, Nepal, India, and some countries from East Africa. Moderate to severe malnutrition was observed across all locations, in contrast to Brazil, which presented a very low percentage of severe malnutrition for all ages. Exposure to risk factors, parasite species, transmission patterns, and access to health services were thought to play a role in the nutritional profiles found by the authors. Here, our AVL patients were from the rural area of Bujaru municipality (Pará State) in the Brazilian Amazon, where 82% of people live below the poverty line (https://www2.mppa.mp.br/sistemas/gcsubsites/upload/53/bujaru(2).pdf, accessed on 12 November 2022). Despite poverty, this population is served by health services (3.50 health units/10,000 inhabitants) and by health agents who visit and monitor the inhabitants, which favored the early diagnosis of AVL patients in the present study, who were immediately treated. Moreover, no case of moderate to severe malnutrition was detected in our casuistic. In this sense, down-regulation of the cellular response to starvation pathway found in symptomatic infection (AVL/SOI profile) was not a surprise.

Unlike previous data that were based on unique expressed genes in each profile, we also performed pathway enrichment analysis by using all up-regulated genes found in each group (Figure 3). We captured shared signatures among the profiles, as well as between the SOI and SRI profiles. Here, SOI profile individuals presented mild signs and symptoms, were not treated with specific antimony therapy, and were followed biweekly for three months period; none of the three individuals have evolved towards to AVL, and all become asymptomatic and presented spontaneous clinical cure during this period, corroborating our previous knowledge about the dynamics of the evolution of human *L.* (*L.*) *chagasi*-infection, which showed that most individuals with SOI profile naturally evolve to the SRI profile, suggesting that they have an intrinsic (genetic) ability to respond positively against to infection [11,12,13,14]. In this direction, we observed a number of shared pathways between the SOI and SRI profiles, in addition to an equivalent activation level, reinforcing a genetic signature of the SOI profile closer to the SRI one than to AVL, which showed a distinct transcriptional profile (Figure 3). Although at a distinct activation level, several pathways were also enriched in both AI profile and AVL concomitantly (Figure 3), showing that the parasite induced shared pathways in distinct clinical–immunological outcomes, regardless the high number of exclusive genes found in each profile.

We also disclosed different levels of activation/repression of genes throughout the profiles (Figure 4) that revealed five distinct transcriptional patterns. In general, activated genes of AI profile were related to innate immune response. Some of the most expressed transcripts encode the chemokine involved in neutrophil activation (CXCL5) [74], in the reactive oxygen species production by phagocytes (NFC4, tetraspanin CD9) [75], and in the recognition of PAMPS on the surface of microbes and orchestration of *TLR*4 functions (CD14) [76]. Conversely, all these genes were down-regulated in AVL (=SI profile). Concerning the intermediate profiles (SRI and SOI) and the indeterminate profile (III), they showed graduated transcription levels between the resistant pole (AI profile) and susceptible AVL (=SI profile).

Altogether, the diverse transcriptional profiles that we detected in the whole spectrum of human *L.* (*L.*) *chagasi* infection subsidized our clinical–immunological approach, which demonstrates the existence of five distinct evolutionary stages of infection caused by this viscerotropic leishmanine parasite in endemic areas. Moreover, this study presented a comprehensive evaluation of blood transcriptional changes that occur after the parasite infection, providing gene signatures that may be useful to better understand AVL immunobiology, as well as the asymptomatic and oligosymptomatic infections by this leishmanine parasite.

## Figures and Tables

**Figure 1 microorganisms-11-00653-f001:**
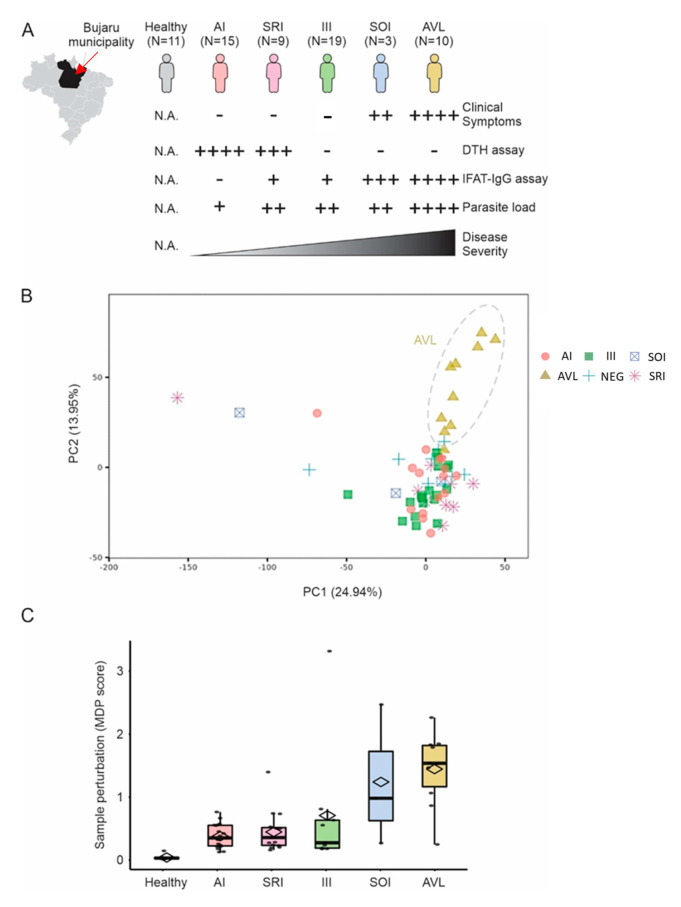
Blood transcriptome of individuals infected with *L.* (*L.*) *chagasi*. (**A**) Blood transcriptome from 67 individuals with AI, SRI, III, SOI, and SI (=AVL) profiles, or without infection (healthy control group), with *L.* (*L.*) *chagasi* from Bujaru municipality, Pará State, Brazilian Amazon, were used in our study. Individuals were grouped according to their clinical–immunological profiles. Clinical symptoms evaluated were: daily fever, weakness, weight loss, hepatosplenomegaly, and hematological changes, such as anemia, leukopenia, and thrombocytopenia. Parasite load was assessed by real-time PCR targeting *Leishmania* kinetoplast minicircles. DTH = delayed-type hypersensitivity assay; IFAT-IgG = indirect fluorescence antibody test. Semi-quantitative scales (+) follow Silveira et al. [12]. (**B**) Principal component analysis using the entire expression data. Colors represent the groups in (**A**). Explained variance is shown for PC1 and PC2 in parenthesis. (**C**) Molecular degree of perturbation (MDP score) of samples. The *y*-axis shows the MDP score using the healthy subjects as a reference group. The scores for each group of infected individuals are shown in the boxplots.

**Figure 2 microorganisms-11-00653-f002:**
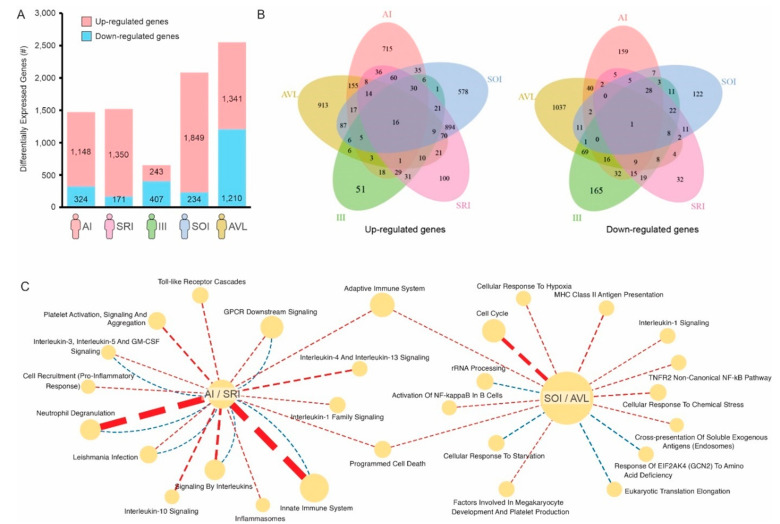
Gene signatures associated with individuals infected with *L.* (*L.*) *chagasi*. (**A**) Number of genes up- (red) or down-regulated (blue) on each group of infected individuals compared to healthy controls. (**B**) Venn diagram showing the number of genes shared by the different groups. (**C**) Reactome pathways specifically associated with individuals with AI/SRI (asymptomatic) profiles or SOI/AVL (symptomatic) profiles. Pathways enriched with up-regulated genes are connected by red dashed lines, whereas pathways enriched with down-regulated genes are connected by blue dashed lines. The thickness of the lines is proportional to the enrichment −log_10_ *p*-value.

**Figure 3 microorganisms-11-00653-f003:**
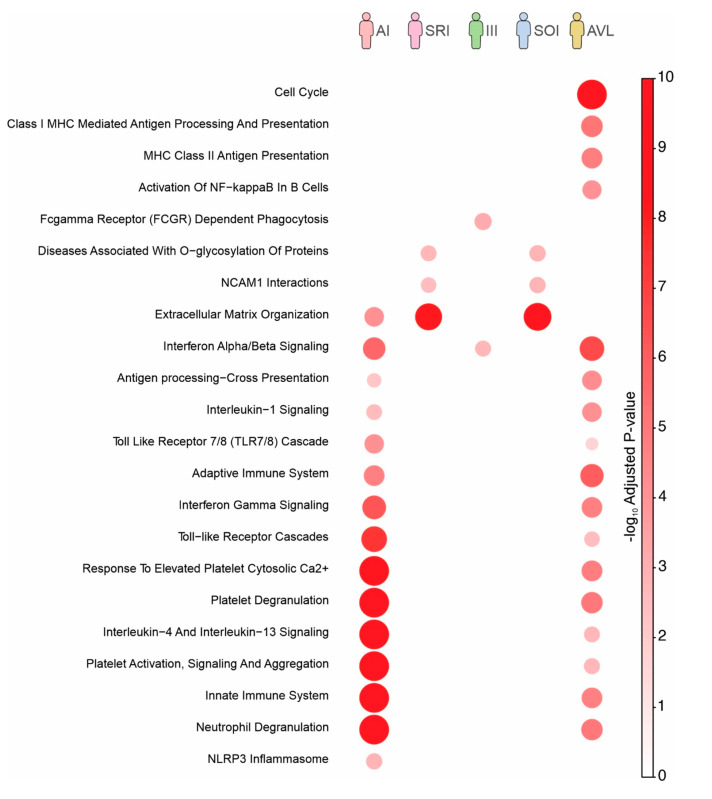
Reactome pathways enriched with genes up-regulated on each infection profile. The color and size of the circles are proportional to the significance of the enrichment.

**Figure 4 microorganisms-11-00653-f004:**
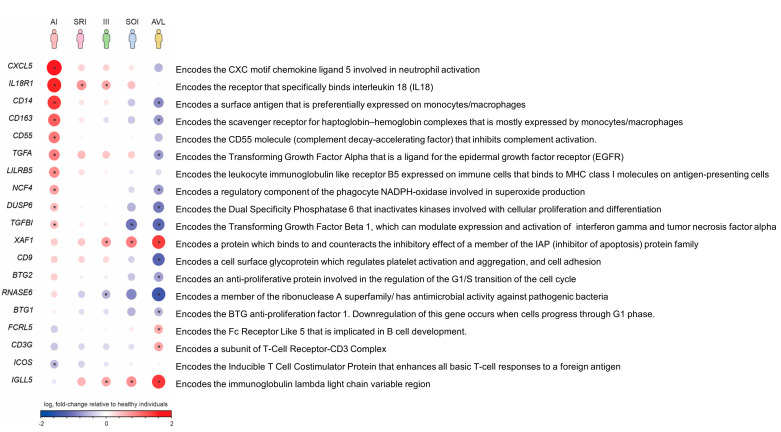
Gene expression shows distinct levels of activation and/or repression throughout the clinical–immunological spectrum of *L.* (*L.*) *chagasi*-infection. Up-regulated genes are shown in red, and down-regulated genes are shown in blue. The color intensity and size of the circles are proportional to the log2FC values (* = FDR < 0.05 in relation to the negative control).

## Data Availability

Data supporting the findings of the present study are available from the corresponding author upon reasonable request.

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
