# Peer review of "Gene Signatures of Symptomatic and Asymptomatic Clinical-Immunological Profiles of Human Infection by Leishmania (L.) chagasi in Amazonian Brazil"

_microorganisms, 2023, doi:10.3390/microorganisms11030653_

Round 1
Reviewer 1 Report
This study attempts to identify the transcriptomic profiles of individuals with different manifestations (symptomatic VS asymptomatic) of L. chagasi infection. The transcriptome analysis is sound and presents interesting novel data, however, while the classification of individuals into 5 different groups (AI, SRI, III, SOI, SI) is clear based on the information provided in Fig 1., throughout the manuscript additional information is provided which appears to confuse the classification scheme and thus makes the discussion rather hard to follow. The authors need to be very careful and very precise in general with the terminology, particularly distinguish clearly between 'clinical symptoms' and 'infection' and please use the descriptive 'clinical-immunological' very carefully--preferably not at all if possible, instead please just stick to the word 'profile', you've defined them clearly enough with the acronyms. In addition, the structure of the manuscript is not ideal, the results section contains statements which belong in the discussion and the discussion section is rather too long and contains information which is more appropriately placed in the methods or even introduction. Please carefully go over the manuscript and try to re-structure it appropriately.
Specific points to clarify and/or revise:
L30-31: sample perturbation or transcriptome perturbation?
L32: greater transcriptomic changes instead of higher
L35: I am not sure that you can claim that there is control of infection, these individuals were not followed up long enough to claim this--all you can say is that the innate immune system is strongly activated in the asymptomatic profiles at the time you chose for the analysis.
L114-115: Please clarify the statements in bald: The former generally evolves into mild clinical signs of susceptibility, with spontaneous clinical infection resolution after one to two months [20]. What are clinical signs of susceptibility and how can you claim that there is clinical infection resolution? Ref. 20 does not state that.
L158: Results section. Please carefully go over your results section, there are several statements in this section which are comments on the implications of the data and they actually belong in the discussion section!
L399-403: You state that AVL patients were diagnosed up to 2 months prior to start of the symptoms and that they were immediately treated. So please clarify when were the patients enrolled in the study? And please clarify was this enrollment prior or post treatment? This information should be moved into the study design, it is not appropriately placed in the discussion.
Author Response
Reviewer 1.
This study attempts to identify the transcriptomic profiles of individuals with different manifestations (symptomatic VS asymptomatic) of L. chagasi infection. The transcriptome analysis is sound and presents interesting novel data, however, while the classification of individuals into 5 different groups (AI, SRI, III, SOI, SI) is clear based on the information provided in Fig 1., throughout the manuscript additional information is provided which appears to confuse the classification scheme and thus makes the discussion rather hard to follow. The authors need to be very careful and very precise in general with the terminology, particularly distinguish clearly between 'clinical symptoms' and 'infection' and please use the descriptive 'clinical-immunological' very carefully--preferably not at all if possible, instead please just stick to the word 'profile', you've defined them clearly enough with the acronyms. In addition, the structure of the manuscript is not ideal, the results section contains statements which belong in the discussion and the discussion section is rather too long and contains information which is more appropriately placed in the methods or even introduction. Please carefully go over the manuscript and try to re-structure it appropriately. First of all, thank you very much for your comments in terms of clarifying the objectives of the manuscript. We have tried, as much as possible, to respond to your suggestions.
Specific points to clarify and/or revise:
L30-31: sample perturbation or transcriptome perturbation?
Transcriptome.
L32: greater transcriptomic changes instead of higher
Ok, already inserted in the text.
L35: I am not sure that you can claim that there is control of infection, these individuals were not followed up long enough to claim this--all you can say is that the innate immune system is strongly activated in the asymptomatic profiles at the time you chose for the analysis.
Sorry, but we have previously demonstrated in cohort studies that the AI ​​and SRI profiles are defined by the presence of the marker of resistance against the infection, the presence of delayed cellular hypersensitivity (DTH+), which is now being shown to have a strong association with the activation of innate immunity and that these immunological factors are associated in controlling the infection. These studies show that no individual who evolved to AVL (=SI profile) previously had a resistance marker (DTH+) against the infection.
L114-115: Please clarify the statements in bald: The former generally evolves into mild clinical signs of susceptibility, with spontaneous clinical infection resolution after one to two months [20]. What are clinical signs of susceptibility and how can you claim that there is clinical infection resolution? Ref. 20 does not state that.
The statements concern the clinical-immunological condition of individuals in the SOI profile who present signs and/or symptoms of mild or moderate intensity, of short duration (from one to two months), and who spontaneously evolve to cure in the referred period; that is, they are individuals who have an incomplete immune-genetic state of susceptibility (Subclinical Oligosymptomatic Infection). Sorry, but the cited reference (Mem Inst Oswaldo Cruz, Rio de Janeiro, Vol. 99(8): 889-893, December 2004) clearly supports what can be read in the statement and it has even been cited before in the same context in another work (Microorganisms 2022, 10, 2188. https://doi.org/10.3390/microorganisms10112188).
L158: Results section. Please carefully go over your results section, there are several statements in this section which are comments on the implications of the data and they actually belong in the discussion section!
You are right!!! We hope that the review we have made meets your suggestion.
L399-403: You state that AVL patients were diagnosed up to 2 months prior to start of the symptoms and that they were immediately treated. So please clarify when were the patients enrolled in the study? And please clarify was this enrollment prior or post treatment? This information should be moved into the study design, it is not appropriately placed in the discussion.
Sorry, you are absolutely right in your comment. In fact, the sentence that is written in the text of the manuscript is incorrect and needs to be corrected as follows: “Despite poverty, this population is served by health services (3.50 health units/10,000 inhabitants) and by health agents who visit and monitor the inhabitants, which favored the early diagnosis of AVL patients in the present study, who were immediately treated”.

Reviewer 2 Report
The article is well written and the topic very interesting.
In particular abstract, keywords, matherial and methods are correctly and representative. Moreover, I suggest to improve in the introduction and discussion a One Health approach by new situations in the future through Remote Sensing approach to improve zoonotic diseases conditions.
For this reasons I suggest you to include these works:
https://doi.org/10.3390/ani12081049
https://doi.org/10.3390/rs12213542
Author Response
Reviewer 2.
The article is well written and the topic very interesting.
In particular abstract, keywords, matherial and methods are correctly and representative. Moreover, I suggest to improve in the introduction and discussion a One Health approach by new situations in the future through Remote Sensing approach to improve zoonotic diseases conditions.
For this reasons I suggest you to include these works:
https://doi.org/10.3390/ani12081049
https://doi.org/10.3390/rs12213542
We are sincerely grateful for the revision you made to our manuscript with your comments and suggestions, especially with regard to the Introduction and Discussion, but in terms of the two suggested works we understand that, at the moment, their approach does not seem compatible with the merely transcriptomic approach to human infection by Leishmania (L.) chagasi that represents the main target of the present manuscript. We hope that changes introduced in the Introduction and Discussion will take in account your comments.
Round 2
Reviewer 1 Report
Dear authors, thank you for incorporating the changes and clarifying the manuscript.